# Impacts of Crystalline Host Rock on Repository Barrier Materials at 250 °C: Hydrothermal Co-Alteration of Wyoming Bentonite and Steel in the Presence of Grimsel Granodiorite

Amber Zandanel [1,*], Kirsten B. Sauer [1], Marlena Rock [1], Florie A. Caporuscio [1], Katherine Telfeyan [1] and Edward N. Matteo [2]

[1]   Los Alamos National Laboratory, Earth and Environmental Sciences Division, Los Alamos, NM 87545, USA
[2]   Sandia National Laboratories, Albuquerque, NM 87185, USA
*   Correspondence: azandanel@lanl.gov

**Abstract:** Direct disposal of dual-purpose canisters (DPC) has been proposed to streamline the disposal of spent nuclear fuel. However, there are scenarios where direct disposal of DPCs may result in temperatures in excess of the specified upper temperature limits for some engineered barrier system (EBS) materials, which may cause alteration within EBS materials dependent on local conditions such as host rock composition, chemistry of the saturating groundwaters, and interactions between barrier materials themselves. Here we report the results of hydrothermal experiments reacting EBS materials—bentonite buffer and steel—with an analogue crystalline host rock and groundwater at 250 °C. Experiment series explored the effect of reaction time on the final products and the effects of the mineral and fluid reactants on different steel types. Post-mortem X-ray diffraction, electron microprobe, and scanning electron microscopy analyses showed characteristic alteration of both bentonite and steel, including the formation of secondary zeolite and calcium silicate hydrate minerals within the bentonite matrix and the formation of iron-bearing clays and metal oxides at the steel surfaces. Swelling clays in the bentonite matrix were not quantitatively altered to non-swelling clay species by the hydrothermal conditions. The combined results of the solution chemistry over time and post-mortem mineralogy suggest that EBS alteration is more sensitive to initial groundwater chemistry than the presence of host rock, where limited potassium concentration in the solution prohibits conversion of the smectite minerals in the bentonite matrix to non-swelling clay species.

**Keywords:** Wyoming bentonite; Grimsel granodiorite; high-level radioactive waste repository; hydrothermal

## 1. Introduction

Several countries have explored the disposal of spent nuclear fuel and high-level nuclear waste in deep geologic repositories contained within a crystalline host rock (e.g., Sweden, Forsmark; Finland, Olkiluoto) and vetted the barrier materials to be used in these repositories up to 150 °C [1–4]. General repository designs consist of a metal waste canister, surrounded by a bentonite clay buffer in a tunnel excavated at depth from crystalline rock. The bentonite buffer serves as a physical and chemical barrier due to its swelling and crack sealing physical properties, and chemical attenuation of the movement of radionuclides in the case of a canister breach [5,6]. Crystalline rock repository concepts have been evaluated in long-term, full-scale, in situ demonstrations at underground research laboratory experiments (e.g., the FEBEX project, Grimsel Test Site, Switzerland [7] and the Äspö Hard Rock Laboratory, Oskarshamn, Sweden [8,9]) that largely specified upper temperature limits for the buffer of 100 °C. Recently, an emphasis on higher temperature studies (>100 °C) has been initiated by both the US DOE and European programs. Specifically, the U.S. Department of Energy's Spent Fuel and Waste Disposition program is conducting generic studies to understand the effect of the higher thermal load associated with the direct disposal of dual-purpose canisters (DPCs) on the EBS system. DPCs are

designed for the storage and transportation of spent nuclear fuel and waste and may contain up to 37 pressurized water reactor (PWR) assemblies [10]. Directly disposing of fuel in DPCs, as opposed to repackaging the waste into waste assemblies with greater fuel spacing, reduces risks and costs associated with spent fuel repackaging [11,12]. While DPC's can be subject to longer intervals of surface storage as part of a thermal management strategy, it is important to understand the potential effects of higher thermal loads in the repository setting, in the cases where longer surface storage is not employed. When shorter intervals of surface storage are employed, the direct disposal of DPCs could result in backfill temperatures greater than 100 °C [11]. Thus, some degree of alteration could occur within the bentonite buffer when exposed to heat generated from the spent fuel package [13,14]. Characterizing the high-temperature interaction of barrier materials with crystalline rock and natural groundwater is therefore important in developing long-term safety and function assessments for a repository system hosted in crystalline rock.

The stability of montmorillonite, the main clay mineral within bentonite, is an extensively researched topic due to its industrial and geologic significance. In the specific application of nuclear waste repositories, previous works have studied a major concern that extensive alteration of montmorillonite would reduce the swelling capacity of the clay barrier and inhibit some of those characteristics that make bentonite a valuable physical and chemical buffer [2,13,15,16]. Montmorillonite alteration is of particular concern where increased temperatures are expected in the repository. Alteration of montmorillonite (a swelling clay) to illite or chlorite (non-swelling clays) is increasingly favorable with increasing temperatures. Heating tests of bentonite materials saturated with crystalline groundwater [17–19] and in situ heating tests in crystalline host rock [4,20] have found decreases to performance characteristics such as cation exchange capacity [1,3,21]. Montmorillonite stability is also highly sensitive to potassium concentrations on the chemistry of saturating solutions, as $K^+$ absorption into smectites may inhibit clay swelling characteristics through collapse of the interlayer space (e.g., [22,23]). Co-alteration of host rock in fluid contact with the bentonite barrier during a heating event may then also be of specific concern, as increased temperatures during a heating event will increase the solubility of potassium-bearing feldspars and micas found in granitic rocks. The influence of crystalline host rock may then affect the engineered barrier systems through ongoing equilibration with any saturating fluids.

Hydrothermal co-alteration of the waste canister materials may additionally affect the stability of the bentonite buffer. For the steel materials being considered in U.S. repository design concepts, the thermal loads and the presence of infiltrating groundwater are anticipated to cause corrosion of steel surfaces [24–27]. Numerous laboratory studies of low-temperature (25–100 °C) hydrothermal alteration have shown the formation of an Fe-oxide layer at steel-bentonite interfaces [28–33]. Co-alteration of steel with other barrier materials may additionally affect both steel corrosion processes and the barrier materials they interact with. The potential for in situ steel alteration in underground repositories was further illustrated by the 18-year FEBEX full-scale EBS heater test at the Grimsel Test Site [20]. In the FEBEX test, bentonite at the heater interface reached temperatures of 30 to 60 °C and corrosion of the carbon steel led to a >140 mm layered zone of iron enrichment extending into the bentonite [4,34], where the iron was interpreted to be hosted in newly formed goethite and sorbed as Fe(II) to clay mineral edge sites [4]. The results of the FEBEX test demonstrate the potential for steel corrosion and Fe mobility into the bentonite buffer in a relatively short time period, even at moderately elevated temperatures. Iron released through steel corrosion may be absorbed by the surrounding clay through cation exchange, reducing the swelling capacity of the clay and by extension the self-sealing properties of the bentonite buffer [35], in a process that may be thermodynamically favorable with increasing temperature [36–38].

Another concern is potential transport of colloids (1–1000 nm suspended particles), which are present in natural groundwaters and can form during bentonite erosion. The dispersion of such colloids may facilitate transport of actinides and other contaminants [39–41].

Colloid-mediated transport of radionuclides is a particular concern for crystalline rock hosted repository systems, where localized connected fracture networks may result in hydraulic conductivity [42–44]. Within a fully saturated bentonite buffer, the hydraulic conductivity is expected to be sufficiently low to prevent transport of radionuclides away from the canister and colloids are expected to be filtered by the low porosity of the bentonite [45]. However, bentonite colloid formation at the bentonite buffer adjacent to the disturbed rock zone of the host rock, and in contact with a fracture or flow pathway, may lead to transport of radionuclides away from the waste package near-field [39,45].

Given the importance of engineered barrier interfaces in the US DOE and international spent fuel programs [46–50], this study investigates hydrothermal interactions between Wyoming bentonite, Grimsel granodiorite, and a synthesized groundwater solution in experiments at pressures and temperatures relevant to high-temperature repository conditions in crystalline rock. A generic crystalline host rock (Grimsel granodiorite) was included as a reactant in addition as the saturating groundwater solution to evaluate the impact of host rock alteration on the solution chemistry and on the stability of the other reactants. Stainless and low-carbon steel were included in the experiments to characterize the potential corrosion of steel waste containers in analogous environments. Reactions were conducted at 250 °C for 6 to 24 weeks, with most experiments conducted over 6 weeks reaction time and three experiments with replicate reactant combinations exploring longer reaction times. Solution samples were extracted weekly throughout each experiment to monitor in situ geochemical changes during the experiments. Mineralogical and geochemical changes were evaluated post-experiment with X-ray diffraction, scanning electron microprobe analyses, and electron microscopy to evaluate alteration of bentonite barrier materials and steel canister materials in crystalline host-rock repository conditions.

## 2. Materials and Methods

Reactions were conducted in flexible gold reaction cells, which were fixed to a 500 mL gasket confined closure reactor and surrounded by an annulus of DI water that controlled pressure within the reaction cell [51]. Solid reactants were combined in the reaction cell and the remaining volume was filled with solution (synthetic Grimsel groundwater), resulting in a range of initial water:rock ratios (WRR) from ~8:1 to 12:1 based on the cell capacity (as illustrated in [52]). WRR decreased during the experimental durations following fluid removal for sampling such that the solution volume in each experiment decreased to as low as ~60% of the initial solution volume (and correspondingly WRR). A high initial WRR was chosen to ensure adequate volume to collect samples as well as to evaluate a highly saturated endmember case. Experiments were heated to 250 °C and maintained at elevated pressure (15 MPa) to simulate an in situ heating event. Experiments ran for between six to twenty-four weeks (Table 1). Three experiments (IEBS-2, -5, and -7) represent replicate experiments with increasing experimental durations and form a sub-series to observe the effect of reaction time on the analyzed results.

Solution samples were collected weekly (5–6 mL per sampling) during the 6- to 8-week experiments and bi-weekly during the 24-week (6-month) experiment. Constant pressure was maintained in the reaction cell by introducing water to the DI water annulus within the confining vessel (see, e.g., [51]); overpressure additionally prevented introducing air into the reaction cell. Samples from the experiments were extracted in airtight syringes. In contact with ambient laboratory conditions the samples were equilibrated to bench conditions (~25 °C, 1 atm) within minutes; precipitation of solid phases was not observed during fluid cooling. An initial aliquot was taken to measure pH, as well as an unfiltered aliquot collected for anion analyses and a filtered (0.22 μm syringe filter) aliquot for an additional cation analyses. All aliquots were stored in polytetrafluoroethylene vials at 1 °C before analysis.

**Table 1.** Initial components and reaction conditions for IEBS experiments in the presence of Grimsel granodiorite. Abbreviations: LCS, low carbon steel; SS, stainless steel; GW, groundwater; GG, Grimsel granodiorite; WB, Wyoming bentonite; EBS, engineered barrier material. Note that IEBS-2, -5, and -7 represent replicate experiments with increasing experimental durations. All experiments were conducted at 250 °C.

| Title | Duration | GW (g) | GG (g) | WB (g) | EBS Type | EBS (g) | Fe (g) | $Fe_3O_4$ (g) | WRR (by Mass) |
|---|---|---|---|---|---|---|---|---|---|
| IEBS-0 | 8 weeks | 160 | - | 16.78 | - | - | 0.5 | 0.5 | 9:1 |
| IEBS-1 | 6 weeks | 144 | 3.47 | 10.91 | - | - | 0.49 | 0.5 | 9:1 |
| IEBS-2 | 6 weeks | 182 | 3.19 | 11.02 | 316 SS | nm | 0.49 | 0.5 | 12:1 |
| IEBS-3 | 6 weeks | 110 | 3.41 | 11.05 | 304 SS | 2.74 | 0.5 | 0.59 | 7:1 |
| IEBS-4 | 6 weeks | 185 | 3.28 | 11.00 | LCS | 5.06 | 0.5 | 0.51 | 12:1 |
| IEBS-5 | 8 weeks | 150 | 3.29 | 11.01 | 316 SS | 5.07 | 0.5 | 0.5 | 9:1 |
| IEBS-7 | 24 weeks | 270 | 6.51 | 21.5 | 316 SS | 5.07 | 0.97 | 0.97 | 9:1 |

nm = not measured.

## 2.1. Materials

Synthetic Grimsel groundwater (GW): the solution used in all experiments was a synthetic groundwater mixed to approximate an average solution chemistry of well samples from the Grimsel Test Site (as reported in [53]). The groundwater at the Grimsel Test Site is a $Na-CO_3$ type water and has a pH of ~8.6 to 8.8. The solution was prepared from double-deionized water and reagent-grade salts. NaOH and HCl were added to adjust the initial solution pH. The resulting solution was then filtered using a 0.45 μm filter to remove undissolved salts, and sparged with He before each experiment to remove oxygen and $CO_2$. The initial synthetic groundwater chemistry is presented in Table 2.

**Table 2.** Initial target solution chemistry of the Grimsel groundwater (GW).

| Parameter | Value |
|---|---|
| pH | 8.4 |
| Ionic strength | 0.005 |

| Constituent | Concentration (mol $L^{-1}$) |
|---|---|
| $Na^+$ | $2 \times 10^{-2}$ |
| $K^+$ | $8 \times 10^{-5}$ |
| $Ca^{2+}$ | $2 \times 10^{-4}$ |
| $Mg^{2+}$ | $5 \times 10^{-4}$ |
| $Cl^-$ | $4 \times 10^{-4}$ |
| $CO_3^{2-}$ | $2 \times 10^{-2}$ |
| Si | $6 \times 10^{-4}$ |
| $SO_4^{2-}$ | $3 \times 10^{-3}$ |

Grimsel granodiorite (GG): all granodiorite samples used in these experiments were sourced from a single drill core from the Grimsel Test Site, crushed and sieved at 2 mm. Grimsel granodiorite used in the experiments comprised 80 wt. % <2 mm and 20 wt. % >2 mm particles. The mineralogy of the unreacted sample included major mineral phases K-feldspar, plagioclase, and quartz, and the minor phases muscovite and biotite. Trace phases allanite, zircon, titanite, and apatite were also found. The bulk mineralogy of representative samples is reported in Table 3.

Wyoming bentonite (WB): bentonite used in this study was provided by Bentonite Performance Minerals LLC from Colony, WY, USA. The WB was pulverized and sieved to <3 mm and used with a free moisture content of ~15.5 wt. % in all experiments. Microprobe analyses of the unreacted bentonite showed it to be dominantly composed of Na-montmorillonite (generalized formula: $Na_{0.33}(Al,Mg)_2(Si_4O_{10})(OH)_2 \cdot nH_2O$), as well as minor clinoptilolite, feldspar, biotite, pyrite, quartz, opal, and sulfide minerals (as characterized by [54]).

**Table 3.** QXRD values of minerals found in analyses of reactants before and after the experiments (see Methods, Section 2.2 for details). Abbreviations: WB = Wyoming bentonite; GG = Grimsel granodiorite. Mineral abundance errors for each phase are approximately ±5 wt. % for clay minerals (smectite, illite, mica, chlorite, kaolinite, saponite) and ±1 wt. % for all other phases.

| Phase | WB | GG | 80 WB:20 GG | IEBS-0 | IEBS-1 | IEBS-2 | IEBS-3 | IEBS-4 | IEBS-5 | IEBS-7 |
|---|---|---|---|---|---|---|---|---|---|---|
| Quartz | 1.5 | 24.1 | 6.9 | 1.3 | 11.4 | 8.2 | 8.2 | 8.2 | 9 | 8.6 |
| K-Feldspar | 0.7 | 10.3 | 3 | 2.1 | 3.5 | 3.7 | 4.2 | 4.4 | 4.4 | 2.3 |
| Plagioclase | 6.2 | 39.3 | 14.1 | 2.3 | 13.6 | 14.8 | 12.7 | 13.9 | 13.4 | 6.1 |
| Apatite | | 0.5 | 0.1 | | 0.4 | 0.4 | 0.3 | 0.2 | 0.4 | |
| Pyrite | 0.2 | 0.3 | 0.2 | | 0.1 | 0.1 | 0.1 | 0.1 | 0.2 | |
| Calcite | | 0.8 | 0.2 | | 0.1 | 0.1 | 0.1 | 0.1 | 0.1 | |
| Dolomite | | 2.3 | 0.6 | | 0.4 | 0.4 | 0.1 | 0.2 | 0.1 | |
| Amphibole | 0.1 | 0.9 | 0.3 | | 0 | 0 | 0 | 0 | 0 | |
| Gypsum | | | | | 0 | 0 | 0 | 0 | 0 | |
| Clinoptilolite | 13 | | 9.9 | 5.2 | 8 | 6.4 | 5.1 | 3.8 | 5.5 | 3 |
| Cristobalite | 1.5 | | 1.1 | 1 | | | | | | 0.6 |
| Buffer | | | | 0.4 | | | | | | 1 |
| Analcime | | | | | | | | | | 2.9 |
| Amorphous + Other | | | | 2.9 | | | | | | 4.2 |
| Smectite + Illite + I/S | 71 | 5.5 | 55.3 | 84.8 | 58.7 | 64.3 | 66.4 | 63.2 | 62.4 | 72 |
| Mica | 3.8 | 14.3 | 6.3 | | 3.4 | 1.5 | 2.5 | 5.7 | 4.3 | 1.5 |
| Chlorite | 2 | 1.8 | 2 | | 0.4 | 0.2 | 0.3 | 0.2 | 0.2 | |
| TOTAL | 100 | 100.1 | 100 | 100 | 100 | 100.1 | 100 | 100 | 100 | 102.2 |

Steel: steel coupons were added to select experiments (Table 1) to simulate the presence of a waste canister. The different steel types used in the experiments included 304 stainless steel (NIST SRM 101g), 316 stainless steel (NIST SRM 160b), and low-carbon steel (LCS: provided by Sandia National Laboratories). The mass of the steel pieces composed ~4–7 wt. % of all reactants.

Redox buffer (iron): redox conditions of all experiments were buffered to a low Eh by adding a 1:1 mixture (by mass) of $Fe_3O_4$ and Fe filings that composed ~0.5% to 0.9% of the total mass of the solid and liquid reactants (~5% of the solid reactants, see Table 1).

## 2.2. Analytical Methods

Aqueous chemistry analyses: the pH (25 °C) at each sampling time point was measured immediately after sampling using a Thermo Orion 4 Star pH probe (Thermo, Waltham, MA, USA). Major cations and trace metals were analyzed via inductively coupled plasma-optical emission spectrometry (ICP-OES) (Optima 2100 DV, Perkin Elmer, Waltham, MA, USA) and inductively coupled plasma-mass spectrometry (ICP-MS) (Elan 6100, Perkin Elmer, Waltham, MA, USA) using EPA methods 200.7 and 200.8 at Los Alamos National Laboratory. Ultra-high purity nitric acid was used in sample and calibration preparation prior to sample analysis. Internal standards (Sc for ICP-OES and Bi, In, and Y for ICP-MS) were added to samples and standards to correct for matrix effects. Standard Reference Material (SRM) 1643e Trace Elements in Water was used to check the accuracy of the multi-element calibrations. Inorganic anion samples were analyzed by ion chromatography (IC) following EPA method 300 on a Dionex DX-600 system (Thermo, Waltham, MA, USA). Typical 2σ uncertainties for the aqueous chemistry results were less than <~5%. In addition, an aliquot of colloidal particles suspended in the solution that remained in the reactor after IEBS-5 was analyzed with a Zetasizer (Los Alamos National Laboratory) to assess particle size and colloid stability.

Geochemical evaluation of aqueous species activities and mineral saturation states were performed with PHREEQC v. 3.5.0 software [55] using the Thermoddem V1.10 geochemical database [56]. Representative mineral solubilities were calculated using a background electrolyte and pH similar to that of the initial GW at 25 °C (Table 2). In situ experimental conditions for the experiments were also calculated from the solution chemistry as analyzed from the anion and filtered cation samples: equilibrium aqueous speciation and in situ pH for each sample were calculated by initially computing aqueous speciation for

the measured solution chemistry of each sample at 25 °C followed by recalculation of pH and solution speciation at experimental conditions (250 °C, 15 MPa). While pressure was included in our calculations for consistency, pressure up to that used in our experiments did not affect the resulting values to the levels of precision included in the presented results. All calculations assumed a starting $HCO_3^-$ concentration of 0.02 mol $L^{-1}$ estimated from the initial mass of $NaHCO_3$ included in the solution.

X-ray diffraction: quantitative X-ray diffraction (QXRD) analyses of the starting material mixture and the bulk reaction products from each experiment are presented in Table 3 in (Supplementary Data, Tables S1–S7). Each sample was ground with 20 wt. % corundum ($Al_2O_3$) for QXRD analysis of the bulk rock [57]. Measurements of the starting materials and IEBS-1 through −5 were conducted with a Siemens D500 diffractometer using Cu-K$\alpha$ radiation. QXRD analyses for IEBS-6, -7, and -0 were performed on a Bruker D8 Advance using Cu-K$\alpha$ radiation. Data were collected from 2 to 70°2θ with a 0.02°2θ step-size and count times of 8 to 12 s per step. Quantitative phase analyses were performed using whole pattern fitting with Jade 9.5 X-ray data evaluation software with the ICDD PDF-4 database and phases are reported as mass%. The average mineral abundance error for each phase is approximately ±1 wt. % for non-clay minerals and ±5 wt. % for clay minerals.

Clay X-ray diffraction (clay XRD): analyses were designed to identify changes to the clay mineral expandability and were conducted on a Bruker D8 Advance using Cu-K$\alpha$ radiation as part of a method. Clay XRD was performed by separating the clay-size fraction (<2 μm) via density separation from a gently crushed portion of the reaction products from each experiment. The Grimsel granodiorite fraction was visually identified and manually removed during the crushing procedure. The XRD patterns of the ethylene glycol-saturated, oriented clay fractions were used to determine alterations to the clay mineral structure through shifts in peak position. Through this method changes in the proportion of non-swelling clays in to swelling clays in the bentonite buffer may be identified.

Scanning electron microscopy: analytical electron microscopy was performed using a FEITM Inspect F scanning electron microscope (SEM) at Los Alamos National Laboratory. All samples were Au/Pd-coated prior to SEM analysis. Imaging with the SEM was performed using a 5.0 kV accelerating voltage and 1.5 spot size. Energy dispersive X-ray spectroscopy (EDS) was performed at 20 kV and a 3.0 spot size.

The reacted steel coupons were embedded in epoxy, then cut and polished to observe a transect of alteration normal to the reacted steel surfaces. Alteration was evaluated by SEM and SEM-EDS of the steel and alteration products at the steel surface. The precipitation thicknesses of the reaction products at the steel surfaces were then measured on backscattered electron (BSE) images of two coupons from each of experiments IEBS-2, -3, -4, -5, and -7. On each coupon, numerous measurements were collected at equal intervals (50 measurements from each long side and eight measurements from each short side). Measurements were made in Adobe Photoshop using the measurement tool. Mineral growth rates were then determined by dividing the average precipitation thickness by the number of experimental run days (see Results, Section 3.3).

Electron microprobe analyses (EMP): EMP analyses were performed at the University of Oklahoma using a Cameca SX50 electron microprobe (CAMECA, Gennevilliers, France) equipped with five wavelength-dispersive spectrometers and a PGT PRISM2000 energy-dispersive X-ray detector. Quantitative analyses were performed using wavelength-dispersive spectrometry with 20 kV accelerating voltage, 20 nA beam current, and 2 μm spot size. Matrix corrections employed the PAP algorithm [58] and oxygen content was calculated by stoichiometry. Counting times were 30 s on peak for all elements; minimum levels of detection (calculated at 3σ above mean background) were within 0.01 to 0.03 wt. % of the oxides for all components except F (0.16 wt. %). All standards for elements in the silicates were analyzed using 30 s count times on the peak, using K-alpha emissions. Typical uncertainties for the EMP analyses were <1% based on long-term standard reproducibility. EMP analyses were made of the clay minerals and granodiorite by encasing the reacted powders

and grains in epoxy before polishing. Measurements were also made of the clay and oxide alteration products identified at the reacted steel surfaces, prepared as described above.

## 3. Results

### 3.1. Aqueous Geochemistry

The pH and concentrations of most analytes in the solution samples achieved a relative steady-state after the first few weeks of experimental time in all experiments. All experiments started at bench pH ~8.5, the pH of the initial synthetic Grimsel groundwater solution. In all experiments, the pH of the fluids initially decreased to below pH 8 before the first sample was collected at week one. After this initial decrease, the in situ pH values in all experiments dominantly remained between pH 7 and 8 (Figure 1a).

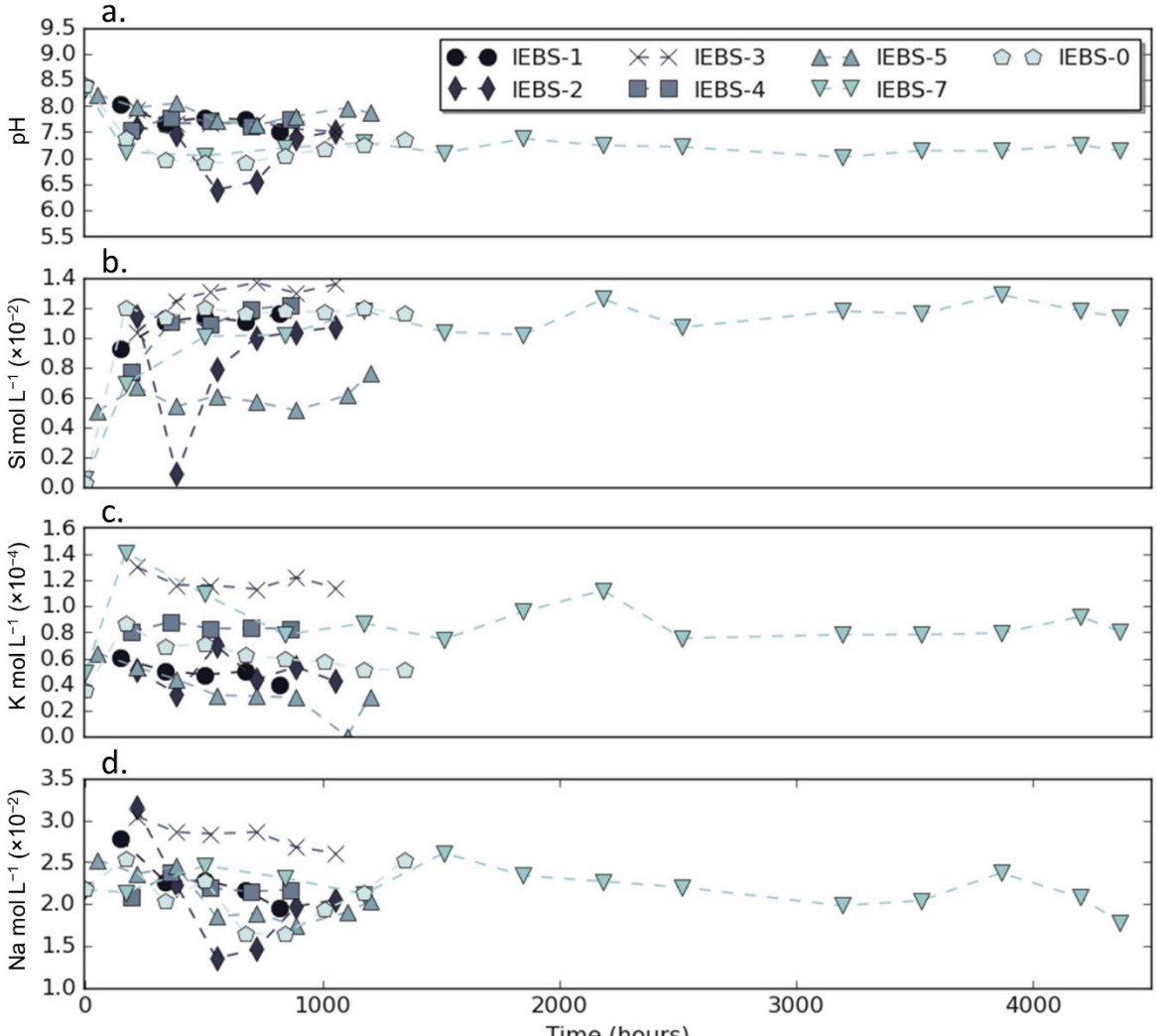

**Figure 1.** Results of fluid sample analyses showing (**a**) In situ pH, (**b**) silica concentrations in mol $L^{-1}$, (**c**) potassium concentrations in mol $L^{-1}$, and (**d**) sodium concentrations in mol $L^{-1}$. Error bars of 1σ standard deviation are within the symbol size.

Silica concentrations ([Si], where concentration in mol $L^{-1}$ is denoted in square brackets for all elements) in all experiments increased rapidly during the first week of experiment time from the starting value (~$5 \times 10^{-4}$ mol $L^{-1}$). Following the initial increases in concentration, the aqueous silica concentrations in the experiments continued to increase at a slower rate or arrive at an apparent steady-state concentration (Figure 1b). This was especially apparent in IEBS-0 (with no GG or steel reactants), which came to an apparent steady-state concentration after the first week of experimental time, compared with all

other experiments that contain GG and continued to have increasing [Si] either throughout the experimental duration (IEBS-3, -4, and -7) or close to the experiment end (IEBS-5).

Potassium (Figure 1c) and sodium (Figure 1d) concentrations typically decreased slightly over the experimental duration. Potassium concentrations initially increased in several experiments (IEBS-0, -3, -4, and -7) before decreasing, and in all experiments remained at values similar in magnitude to those observed in the starting solution. No paired [K] decrease with [Na] increase was identified, in contrast to aqueous chemistries of alteration experiments with initially higher (~100×) potassium concentrations (i.e., [54]). Calcium concentrations typically decreased continuously following the initial sample, including throughout the 24-week experiment (IEBS-7). Iron concentrations in all experiments remain below ~$2 \times 10^{-5}$ mol L$^{-1}$ for the duration of the experiments. Aluminum concentrations increased to ~$1$–$2 \times 10^{-4}$ mol L$^{-1}$ by the first weeks of experiment time and remained near constant or slightly decreased for the remainder of the experiments; quench samples (collected after experiment cooling) from IEBS-2 and IEBS-3 had sharp increases in concentration of both [Fe] and [Al].

Sulfate concentrations in all experiments increased from the initial GG solution concentration (~$3 \times 10^{-3}$ mol L$^{-1}$) to up to $1 \times 10^{-2}$ mol L$^{-1}$ by the first sample taken after ~1 week of experimental time. Sulfur concentrations in solution then remained between $5 \times 10^{-3}$ and $1 \times 10^{-2}$ mol L$^{-1}$ and generally stabilized to a steady-state concentration by the fourth week. Complete aqueous chemistry results are included as Supplementary Data (Tables S1–S7).

*3.2. Mineralogy*

QXRD analyses showed trends of mineral dissolution, precipitation, and recrystallization reactions that occurred during the experiments. Post reaction analyses of IEBS-0 can be compared to the initial analysis of the unreacted WB; post-reaction analyses of IEBS-1 through -5 and IEBS-7 are more directly comparable to XRD analysis of the unreacted mixture of WB and GG as used in those experiments (80% WB to 20% GG, Table 3). In the baseline experiment that reacted Wyoming bentonite with the synthetic Grimsel groundwater solution (IEBS-0), QXRD analysis of the reacted clay matrix showed relative reductions in plagioclase feldspar and clinoptilolite and an increase in smectite (as Smectite + Illite + I/S, Table 3). In all experiments IEBS-1 through IEBS-5, the relative abundances of plagioclase feldspar, clinoptilolite, micas, and chlorite decreased, while the relative abundances of quartz and smectite increased in comparison to the analyzed 80% WB to 20% GG starting mix. Relative increases especially of quartz are interpreted to be at least in part attributable to the dissolution, and thus relative decrease, of the accessory minerals plagioclase and clinoptilolite (as well as the reduction or disappearance of minor phases pyrite, apatite, calcite/dolomite, and amphibole). In the 24-week experiment (IEBS-7) we observed greater bulk mineralogical changes in the QXRD results than the shorter-term experiments with the same reactants (IEBS-2 and IEBS-5), including larger decreases in the relative abundances of plagioclase feldspar, K-feldspar, and clinoptilolite and a greater increase in the abundance of smectite (from ~55 wt. % to 72 wt. %). Formation of secondary zeolites (as analcime) were also detected in IEBS-7 (Table 3, Figure 2).

The clay XRD analyses of the ethylene glycol saturated, oriented clay fractions had peak positions from all experiments that were similar to those of the unreacted Wyoming bentonite. For example, observed glycolated smectite (001) distances were all between 16.9 and 17.1 for reacted samples, in comparison to 17.0 for unreacted bentonite. The differences between d (002) and d (003) spacings ranged from 5.1 to 5.3 for all reacted samples (5.2 for unreacted samples). Therefore, significant illitization, interlayered illite-smectite formation, or montmorillonite structural changes likely did not occur as montmorillonite peak positions from the heated samples did not show appreciable shifts in comparison to unreacted bentonite.

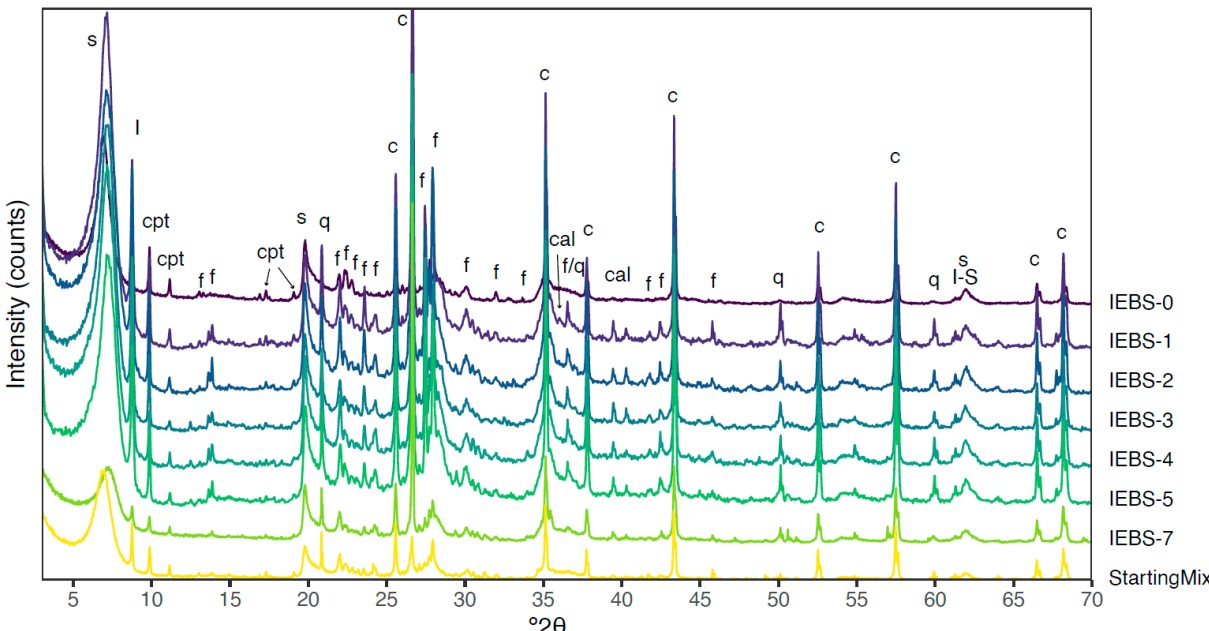

**Figure 2.** QXRD patterns of the Starting Mix of reactants (20% GG: 80% WB) and of the reacted mixture of solid phases from each experiment. Peaks identified and labelled include smectite (s), illite (I), clinoptilolite (cpt), feldspars (f), quartz (q), calcite (cal), mixed illite-smectite (I–S), and corundum (c, used as internal standard). Note that few differences in peaks are visually identifiable between different samples, illustrating the lack of substantive alteration of the mineral reactants. Quantitative analyses were performed as described in the Methods to better identify minor changes.

SEM imaging of loose powder mounts of the reaction products in all experiments, including IEBS-0 reacting Wyoming bentonite with groundwater and no other reactants, showed the development of a foily texture in the fine-grained clay matrix (Figure 3a). No recrystallization of montmorillonite to non-swelling phases, such as illite or muscovite, was identified. EMP analyses of the clay matrix indicated that the chemical compositions of the reactants from the IEBS experiments and the starting products had similar major element compositions (~60 wt. % $SiO_2$, ~22 wt. % $Al_2O_3$, 4 to 6 wt. % FeO and 1% to 2% MgO, and 1% to 3% of $Na_2O$, ~0.3 $K_2O$, 0.2 to 0.5 wt. % CaO, and 0.1 to 0.2 wt. % F).

In all the experiments with Grimsel granodiorite and Wyoming bentonite (IEBS-1 to IEBS-5 and IEBS-7), spherical calcium silicate hydrate (CSH) or calcium (alumino-)silicate hydrate (C(A)SH) phases were observed by SEM and thin section to have formed within the fine-grained clay matrix. Small amounts of this mineral were observed in IEBS-1 and were abundant in samples collected after IEBS-2 through 5 (Figure 3c,d). Results of EMP analyses were consistent with a CSH such as tobermorite ($Ca_5Si_6O_{16}(OH)_2$ $4H_2O$). However, no such phases were identified via XRD, indicating low bulk abundance or potentially a poorly crystalline nature.

Secondary zeolites were observed in several experiments. Isolated zeolites were observed in IEBS-0 (Figure 3b), and SEM images showed abundant zeolites in thin sections prepared from smectite matrix reacted in IEBS-7 (24-week experiment). BSE and microprobe investigation of the crystals indicated chemical ratios consistent with analcime ($NaAlSi_2O_6$ $H_2O$) with additional Ca, indicating formation consistent with the solid-solution relationship between analcime and wairakite ($CaAl_2Si_4O_{12}$ $2H_2O$). Gypsum crystals and Fe-smectite were additionally identified embedded in the smectite matrix in IEBS-2.

Step textures evidencing dissolution were observed on feldspar grains in the reacted Grimsel granodiorite on the exterior surfaces that were exposed to water-rock interaction (see step textures of an albite grain in Figure 3e). Evidence of graniodiorite dissolution was also observed in thin sections prepared from granodiorite samples recovered after

experiments that exhibited void spaces, especially around K-feldspar (Figure 3f) and plagioclase grains.

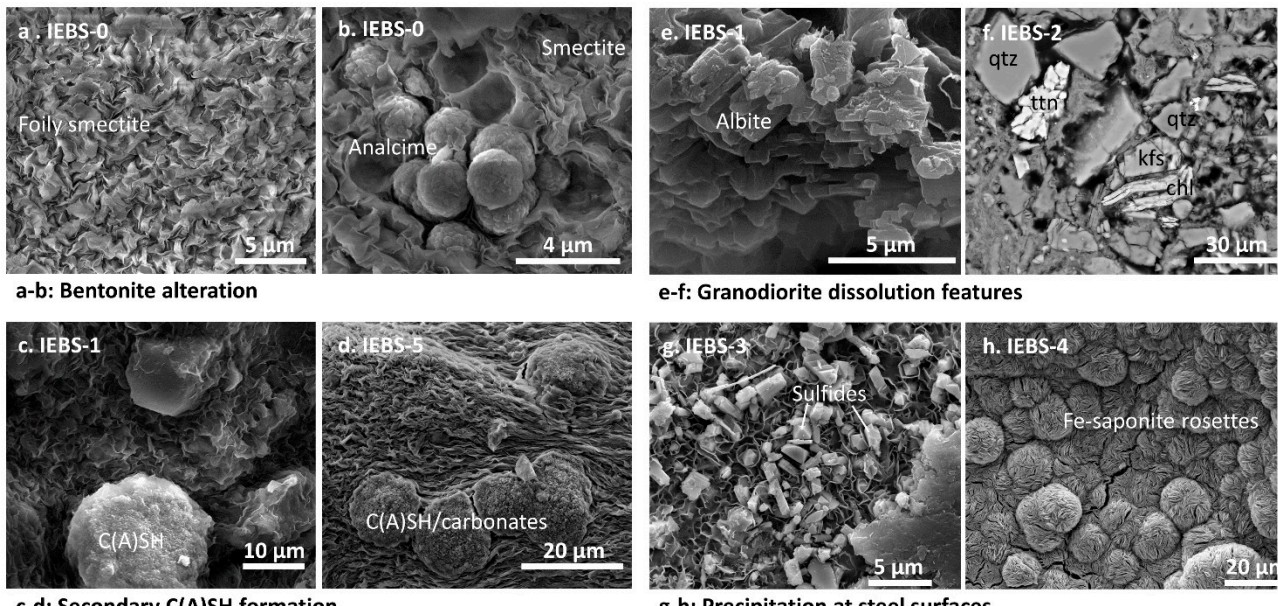

**Figure 3.** SE images of post-experimental solid reactants. (**a**) Characteristic foily texture of smectite reacted in GG. (**b**) CSH crystals embedded in the post-IEBS-0 smectite (Wyoming bentonite) clay matrix. (**c**) Secondary C(A)SH formed in the smectite matrix reacted in IEBS-1. (**d**) Secondary C(A)SH/carbonate phase embedded in the dried montmorillonite gel recovered from IEBS-5. (**e**) A granodiorite fragment after reaction in IEBS-4, with dissolution features (step texture). (**f**) Thin section of Grimsel granodiorite fragment after reaction in IEBS-2; note the void spaces between grains which may be related to partial dissolution of feldspar grains. (**g**) Fe-saponite (honeycomb texture) with embedded Fe,Ni,Cr-sulfide minerals (light gray crystals) on the reacted 304 SS surface. (**h**) Fe-saponite rosette at the LCS surface after reaction in IEBS-4.

### 3.3. Steel Coupons

The steel coupons (316 SS, 304 SS, and LCS) reacted in experiments IEBS-2 through IEBS-5 and IEBS-7 resulted in a distinctive secondary mineral assemblage formed at the steel surfaces that was not widely evident in the smectite matrix or at the reacted granodiorite surfaces (Figure 4a). All reacted types of steel (316 SS, 304 SS, and low-carbon steel) showed the characteristic growth of abundant Fe-smectite identified as Fe-saponite by EMP at the steel surface (Figure 4b). IEBS-2, -5, and -7 included 316 SS. We identified two layers of mineral growth that formed perpendicular to the steel surface, where chromite or Fe,Cr,Ni-oxides were observed to form a thin layer locally adjacent to the 316 SS surface and a layer of Fe-saponite formed directly adjacent to the localized chromite deposits and to the pitted 316 SS surface (Figure 4a). Clay minerals more distal to the steel surface had much lower Fe-content, comparable to that of the bulk clay matrix (Figure 4b). Fe,Cr,Ni sulfides were also observed on the 316 SS reacted in IEBS-2. Reaction products observed on the surface of the 316 SS from IEBS-7 also included localized coatings of a CSH/smectite phase formed within and on a mat of Fe-saponite rosettes. The surface of the post-reaction 304 SS coupon (IEBS-3) had Fe-Ni-Cr sulfides embedded in honeycomb-textured Fe-saponite (Figure 3g) and a layer of Fe-poor smectite was observed attached to the underlying Fe-saponite, distal to the steel surface. The post-reaction LCS (IEBS-4) was coated by a layer of Fe-saponite rosettes: no chromite, [Fe, Ni, Cr]-sulfides, or other Fe-rich alteration products were identified (Figure 3h).

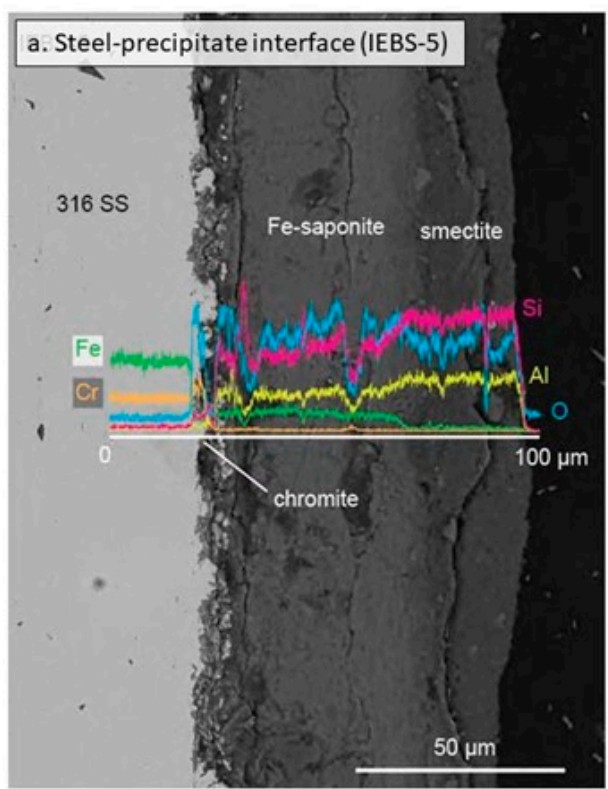
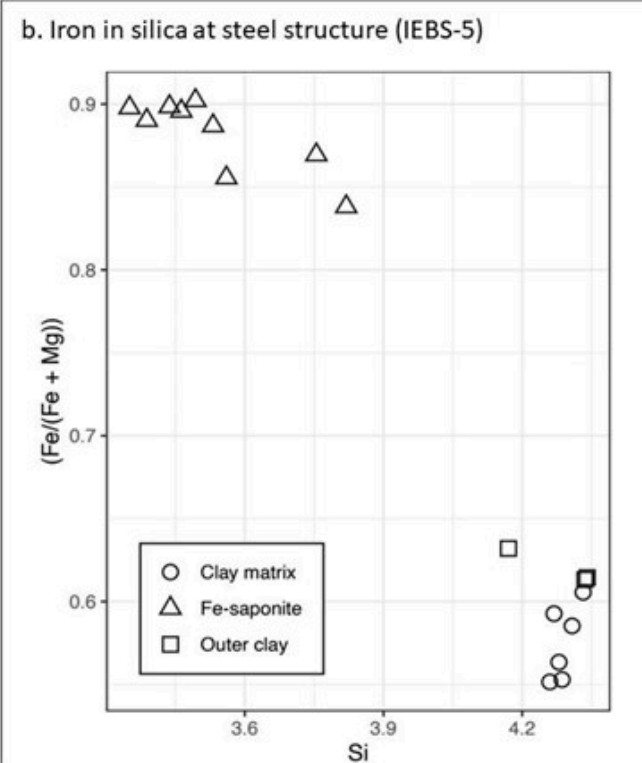

**Figure 4.** Alteration at the steel-secondary mineral interface. (**a**) Energy dispersive X-ray spectroscopy (EDS) chemical results from a line scan (white line) across the steel-clay boundary from IEBS-5 (GG + WB + 316 SS). Relative elemental abundances of Fe, Cr, Si, Al, and O are identified by color and label. (**b**) Selected EMP results from clay at the steel surface from IEBS-5. Outer clay refers to the clay outboard of the Fe-enriched zone on the steel coupon (labeled as smectite in (**a**)). Clay matrix refers to the composition of the clay matrix that makes up the reaction products not attached to the steel.

Measurements of the thickness of the layer of precipitated secondary minerals at the steel surfaces (Table 4) showed the greatest amount of precipitation (by greatest linear thickness of the precipitation growth measured from the steel surface) occurred in the 24-week experiment (IEBS-7). Normalized to the experimental time, the fastest precipitation rates were those at the surface of the LCS coupon (IEBS-4, 1.12 $\mu$m day$^{-1}$), followed by the 304 SS (IEBS-3, 0.88 $\mu$m day$^{-1}$). The calculated precipitation rates were the slowest at the 316 SS surfaces. The six-week experiment with 316 SS, IEBS-2, had a lower precipitation rate (0.06 $\mu$m day$^{-1}$) versus the 8-week experiment, IEBS-5, at the same conditions (0.69 $\mu$m day$^{-1}$). However, due to the extremely heterogeneous thicknesses of the secondary precipitate observed (see standard deviations listed in Table 4), we interpret the apparent differences between both precipitation thickness and rates as being within the uncertainty caused by the areal variability of the secondary products.

**Table 4.** Secondary precipitation thicknesses and growth rates at steel surfaces. SS = stainless steel; LCS = low-carbon steel.

| Expt. | Expt. Duration | Steel Type | Average Precipitation Thickness ($\mu$m) | Precipitation Rate ($\mu$m Day$^{-1}$) |
|---|---|---|---|---|
| IEBS-3 | 6 weeks | 304 SS | 31.60 ($\pm$27.01) | 0.88 |
| IEBS-2 | 6 weeks | 316 SS | 2.27 ($\pm$1.40) | 0.06 |
| IEBS-5 | 8 weeks | 316 SS | 38.72 ($\pm$27.76) | 0.69 |
| IEBS-7 | 24 weeks | 316 SS | 45.94 ($\pm$23.58) | 0.25 |
| IEBS-4 | 6 weeks | LCS | 40.17 ($\pm$30.17) | 1.12 |

### 3.4. Colloid Formation

A gel phase was observed on experiment cooling in IEBS-3, IEBS-4, and IEBS-5 (Figure 5). When suspended in DI water, particles remained suspended in solution. A dried film of the suspension from IEBS-5 was analyzed via XRD and was identified as montmorillonite. Imaging the dried gel in SEM showed that the dried gel texture consisted of both linear and cross-linked morphologies (Figure 5a–c). The measured zeta potential values centered around −38.9, indicating moderately stable colloids. The average particle size diameter was ~237 nm and the diameters were distributed between ~30 and 1000 nm (Figure 5d).

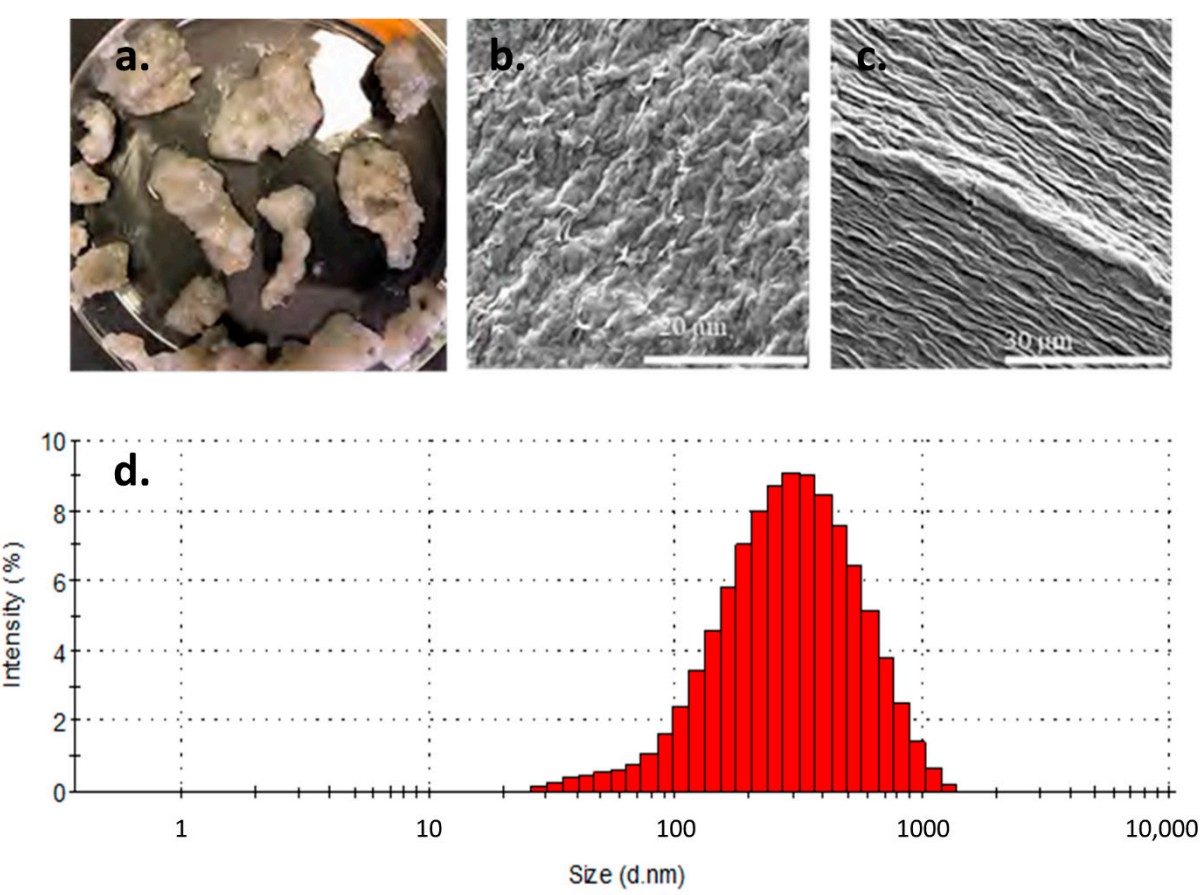

**Figure 5.** (**a**) Post-reaction gel mixed with clay from IEBS-4. (**b**) SEM images of the dried gel with cross-linked texture. (**c**) Dried gel with linear texture. (**d**) Distribution of colloid diameters (nm) from sample IEBS-5, dispersed in DI water and measured via Zetasizer.

## 4. Discussion

The GG and GW of these experiments were selected to be representative of a crystalline host rock and a groundwater that had equilibrated with that rock such that it was at a steady-state fluid chemistry with respect to its mineral assemblage at an in situ temperature considerably lower than the temperatures that could be reached in a heating event. Alteration of the more reactive minerals present in the GG was then expected to occur at lab timescales as the GG and GW reacted to the elevated experimental temperature. This alteration was evidenced by the dissolution features and mineralogical changes observed to the reacted GG: secondary phases formed in the IEBS experiments include a fine-grained, recrystallized clay matrix with phenocrysts derived from the starting granodiorite and accessory minerals in Wyoming bentonite. The effect of GG alteration on the barrier materials was also observed in the fluid phase, as evidenced by silica concentration trends in solution: [Si] in IEBS-0 that included no GG as a reactant came to an apparent steady-state concentration after the first week of experimental time, as compared with all other experiments that contain GG and continued to have increasing [Si] throughout their experiment

durations. The increasing [Si] in the experiments indicates ongoing dissolution of minerals either within the GG, or potentially alteration of the WB caused by equilibration of GG with the GW solution at the experimental temperature. Fluid chemistry trends and final reaction products were then monitored to: (1) evaluate the stability of the barrier materials WB and steel in the presence of crystalline host rock and groundwater, and (2) understand the stability of specific secondary mineral assemblages that may form in the investigated repository conditions.

### 4.1. Mineral Solubilities

Solution species that underwent an initially rapid increase or decrease in concentration before approaching quasi-steady state concentrations can give insight into the reactive minerals and initial mineralogical changes occurring in situ. Specifically, the initial increase in [Si] and decrease in [Ca] indicates an initial equilibration to Si- and Ca-bearing solid phases potentially resulting from dissolution or precipitation. Figure 6 illustrates changing Si and Ca solubilities for select minerals identified in the initial reactants (quartz ($SiO_2$), smectite, and calcite) and secondary minerals identified to have formed in situ during the experiments (tobermorite, analcime, and wairakite). During an in situ heating event in an underground repository, high temperatures (>~100 °C) may increase the solubility of smectite relative to zeolite minerals (Figure 6a). Retrograde solubility of CSH minerals will also increasingly favor the formation of tobermorite and similar phases with increasing temperature (Figure 6a,b). This is consistent with the results of our experiments of zeolite (analcime) and CSH formation from the alteration of WB including smectite and accessory minerals.

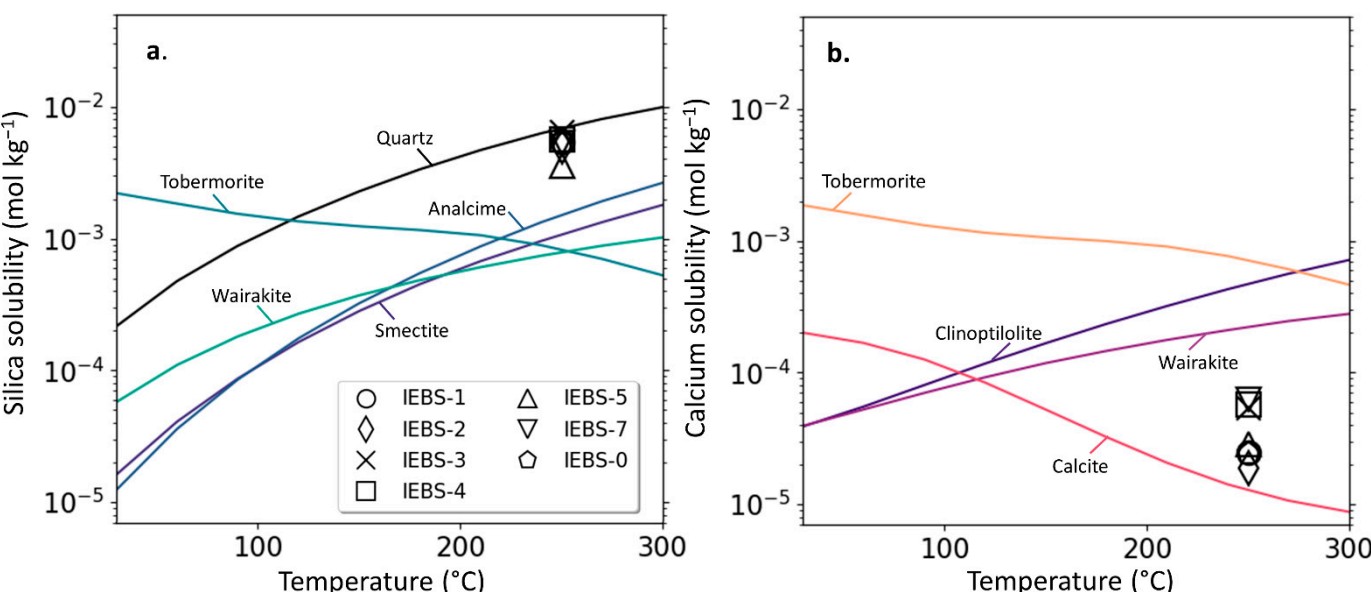

**Figure 6.** Silica and calcium solubility in solution with respect to select minerals identified in the initial and/or post-mortem reactants, at pH 7.75 and 15 MPa. (**a**). Silica solubility for select minerals and silica concentrations for samples collected at ~1000 h from each experiment, and (**b**). calcium solubility for select minerals and calcium concentrations from the same samples plotted in (**a**).

After the first 2 to 3 weeks of experiment time, [Si] in most experiments appeared to achieve a quasi-steady-state concentration and the rate of [Ca] decrease had slowed so that the decrease throughout the remaining experiment time was not substantial. Samples taken at six weeks (~1000 h) of experiment time can then be compared to Si and Ca solubilities to evaluate the relative stability of specific minerals at their respective quasi-steady-state concentrations at the selected pressure and temperatures. In all experiments, [Si] is grouped around ~5 × $10^{-3}$ mol L$^{-1}$—below the Si solubility with respect to quartz, and greater than concentrations calculated to be in equilibrium with smectite as well as secondary CSH

and zeolite minerals. From the QXRD data that show decreasing wt. % of clinoptilolite and the formation of analcime only over longer reaction times (24 weeks, Table 3) this may suggest that the rate of dissolution of clinoptilolite exceeds the rate of formation of CSH and zeolite minerals from smectite alteration at the experimental conditions.

The formation of CSH minerals and the relatively insoluble wairakite may additionally be inhibited by low $Ca^{2+}$ activity in solution. After the initial decrease in calcium concentrations, potentially through formation of CSH and carbonate phases (Figure 3c,d) and/or Ca-bearing analcime, calcium concentrations in our experiments were below the solubility values for the selected zeolites and tobermorite. The low calcite solubility and $NaHCO_3$ solution of the GW may control [Ca] at a low concentration relative to the solubility of the observed secondary minerals under the experimental conditions, thus inhibiting formation of these phases. This is consistent with the observed increase in abundance of the Na-zeolite analcime after longer reaction times in our experiments, and the lack of identifiable increase in the mass of secondary CSH minerals over time. In an underground repository with $CO_3$-bearing groundwaters, increases in temperature and pH values that decrease calcite solubility would then be expected to suppress the formation of CSH minerals or Ca-bearing zeolites in the long term by favoring calcite stability and relatively low [Ca].

### 4.2. Clay Alteration

QXRD analyses of the solid reactants results showed that the smectite-illite fraction increased compared to the non-clay fraction, at least in part attributable to relative decreases in the accessory minerals in the WB (specifically clinoptilolite, pyrite, amphibole, and plagioclase, Table 2), as well as natural variability. Results of clay XRD analyses designed to. Similarly, EMP results did not indicate increases in [K] or decreases in [Na] within the bentonite after alteration. Previous studies have shown that illitization may occur at circumneutral to high pH and high temperatures: for example, ref. [59] showed that under pH conditions of 11 to 13, Na-rectorite was formed at 150 to 200 °C within 17 days and Na-mica (paragonite) developed after 32 days. Cheshire et al. [54] observed significant decrease in [K] and correlated increases in [Na], interpreted as driven by $K^+$ exchange in the smectite. In contrast, the initial [K] in the GW in our experiments was more than two orders of magnitude lower than comparable hydrothermal experiments that observed smectite illitization during the experiments [54].

In a granodiorite-hosted system, equilibration of saturating groundwater solutions with potassium-bearing host rock may be an additional source of potassium to the bentonite barrier. Analyses of the reacted granodiorite through SEM and XRD analyses showed in situ dissolution of K-feldspar (Figure 3) and mica (Table 3) indicating potential contribution of $K^+$ to the reactant solution. However, increases in potassium concentrations were not observed throughout the experimental durations. While this lack of consistent or ongoing [K] increase in solution suggests $K^+$ sequestration potentially through exchange in the montmorillonite, the proportion of non-swelling clay minerals within the clay phase reacted in experiments that included GG were not substantially increased from either the initial WB or from the reacted clay from the experiment excluding GG (IEBS-0: see Section 3). This has long-term implications for bentonite stability in a crystalline host rock in that K-poor systems might be expected to preserve the overall swelling properties and cation exchange capacity of bentonite buffer materials, whereas K-rich systems may pose a complicating factor to the long-term efficacy of buffer materials [60,61]. Our results did not show evidence of illitization or increases in other non-swelling clay species in bentonite buffer material during a heating event, despite the high temperatures (250 °C), alteration of potassium-bearing minerals in the crystalline host rock, and relatively long reaction time for the 24-week experiment.

### 4.3. Secondary Calcium (Alumino-)Silicate Hydrates and Zeolite Minerals

CSH phases (identified in places as tobermorite) were observed embedded within the fine-grained clay matrix after all experiments. Because of their location and presence

in the WB-only experiment (IEBS-0), these phases were interpreted to have formed as an alteration product of smectite and/or the accessory minerals found in the WB (Table 3). This is consistent with tobermorite formation previously identified in clay-rich natural [62] and experimental [63] hydrothermal systems. In experimental systems, tobermorite forms due to smectite alteration, usually at alkaline bulk chemistries (pH > ~10 [63–65] and at cement-bentonite contacts in hydrothermal conditions up to 200 °C [3,66,67].

The CSH minerals identified in our experiments notably formed in conditions with a relatively low-pH bulk fluid chemistry. The formation of CSH (and C(A)SH) minerals is known to be sensitive to the pH of the solution. Ref. [68] describe the formation of tobermorite with the generalized reaction:

$$Ca_2^+ + SiO_2\ (aq) + H_2O \rightarrow tobermorite + 2H^+ \tag{1}$$

in which $H^+$ is produced and CSH mineral formation is favored at pH > 11.5 [69]. All the IEBS experiments in this study had substantially lower in situ pH (<8) throughout their experimental durations. Similarly, the mineral saturation calculations from the bulk solution of the experiments where CSH minerals formed also indicate a thermodynamically unfavorable environment for precipitation. For example, the low calcium concentrations in the bulk solution from all experiments were below the calcium concentration considered at equilibrium saturation with tobermorite (Figure 6b). This strongly suggests that even at the high water-rock ratios, local fluid chemistries at the mineral-solution interface were not reflected in the bulk solution. Local conversions of primary minerals (smectite) to secondary products (including CSH phases) may then be best identified through direct observation of the mineralogy rather than by samples of bulk solutions or groundwater, based on available low-temperature stability data for CSH minerals extrapolated to in situ conditions. However, we note that stability data for CSH minerals at high temperatures (>150 °C) are currently sparse and may be substantially improved by additional empirical studies. In contrast to CSH minerals that may be more likely to form where pH > 11.5, zeolite formation following bentonite alteration is favorable in lower-pH solutions [69,70]. In our experiments, secondary zeolites (analcime) were quantitatively identified following the 24-week experiment (IEBS-7, Table 3). As IEBS-7 is a longer-duration replicate of experiments IEBS-2 and IEBS-5, having the same combination of initial reactants, this may indicate that secondary zeolite formation rate progresses slowly during at least the first eight weeks of reaction time at 250 °C. This is complementary to similar six- to eight-week hydrothermal experiments WB ± argillite host rock (Opalinus clay) in a solution with circumneutral pH, where zeolite phases (analcime–wairakite solid solution) formed as a dominant secondary mineral at 300 °C but were not identified in identical experiments conducted at 200 °C [30,52]. Another zeolite mineral, clinoptilolite, was identified as a constituent of the starting WB mineral assemblage but the relative percent of this zeolite decreased after all experiments (Table 3) and is not interpreted to have formed in situ.

### 4.4. Steel-Bentonite Interface Reactions

The results from these experiments show a dynamic environment in the experimental systems at the bentonite-metal interface. The similarities in mineral precipitation at the steel surface between experiments suggest that the bulk chemistry, rather than differences in steel type, likely controls the alteration mineralogy. The post-experiment steel surfaces from all experiments showed uniform corrosion and alteration products that are categorized here as four different layers: (1) general corrosion of the steel surface; (2) Fe-oxides and other metal oxides (such as chromite) directly proximal to the steel surface (Figure 4); (3) CSH and/or chlorite (if present); and (4) Fe-smectite (often as Fe-saponite) outer layer with occasional Fe,Ni,Cr-sulfide deposits (Figure 3g), similar to the layered precipitate sequences described forming at the steel surfaces by [30].

Fe-saponite formation is related to the interaction of the Fe-bearing/Si-rich fluids from the leaching of the steel and bentonite dissolution [54,71]. Synthetic Fe-saponites are known to crystallize in dilute solutions and gels of silica, Fe-, Al- chlorides at temperatures

up to 850 °C [72]. This may be analogous to the local environment during dissolution or partial dissolution of the steel plates that contributes ferrous iron into a fluid phase by the following reactions [73]:

$$4\,Fe^o + 3\,O_2 \rightarrow 2\,Fe_2O_3 \tag{2}$$

$$2\,Fe + 2\,H_2O + O_2 \rightarrow 2\,Fe(OH)_2 \tag{3}$$

where steel dissolution occurs in a solution containing silica and aluminum, this reaction may then facilitate Fe-saponite (smectite) crystallization with the steel surfaces acting as a growth substrate. The rate of alteration of the bentonite in proximity to the steel corrosion is then dependent on the ability of the iron to migrate through the clay as $Fe^{2+}$. This is regulated by the rate of corrosion, the rate of formation of the Fe-oxide, and the system dynamics [74]. Previous studies have also characterized Fe-saponite alteration into chlorite in the presence of ferrous iron at temperatures approaching 300 °C and near-neutral pH [74]. These results were further confirmed to apply to long-term systems by [75] through long duration experiments (up to 9 years). The authors demonstrated that smectite is consumed by dissolution to produce chlorite (chamosite) by precipitation by the following reaction:

$$3\,smectite + 3\,Fe + 4\,H_2O \rightarrow 1\,chlorite + 3\,quartz + 2\,albite + 3\,H_2 + zeolite \tag{4}$$

The Fe-enriched phyllosilicate and sulfate minerals observed were described by the interaction between iron supplied by steel corrosion and smectite through the following reactions [30]:

$$Fe^{2+} + Ni^{2+} + Cr^{3+} + H_2S\,(aq) + \underset{\text{smectite}}{(Na,K,Ca)_{0.33}(Al_{1.67},Fe_{0.2},Mg_{0.13})Si_4O_{10}(OH)_2}$$
$$\rightarrow \underset{\text{pentlandite}}{(Fe,Ni,Cr)_9S_8} + \underset{\text{Fe-saponite}}{(Na,\,K,\,Ca)_{0.33}Fe_3(Si_{3.67},Al_{0.33})O_{10}(OH)_2} \tag{5}$$

In our experiments with 316 SS (IEBS-2, IEBS-5, and IEBS-7), the location, chemistry, and morphology of the clays at the steel surface is consistent with a local iron-rich chemistry that fosters the reactions described above. The occurrence of Fe-saponite closely associated with the steel surfaces, with no significant Fe-content found in the smectite more distal from the steel surface (>100 µm), indicates that the formation of the Fe-smectite is closely related to the local availability of Fe from the steel corrosion processes. The morphology and attachment of the surface-bound minerals on the steel were interpreted as evidence that these minerals directly precipitated in the localized environments surrounding the metal, with the steel material acting as a substrate for mineral growth in response to corrosion. The presence of the newly formed Fe-rich phases together with the lack of significant increase in aqueous iron in the solution indicate that the localized mineralogical reactions at the steel surfaces did not influence the bulk solution chemistry.

*4.5. Colloid Formation*

The formation of colloidal bentonite gel phase visible by naked eye in IEBS-3, -4, and -5 is of particular interest in evaluating the potential for colloid-mediated transport in the presence of a crystalline host rock. As the gel was observed after experiment termination and at room temperature, it is not known whether aggregation occurred at experimental conditions or during cooling. The stability of bentonite colloids at the experimental temperature (250 °C) is not well studied: previous studies of montmorillonite colloid formation in relevant hydrothermal conditions have focused on temperatures <100 °C [53,76–79], while at higher temperatures the much lower dielectric constant of water as well as increased smectite solubility (Figure 5) may have unexplored effects on colloid stability. The presence of an aggregated gel at bench temperatures in our quenched experiments, however, suggests that even if colloids are stable in solution at the elevated temperatures next to a heating source, transport may still be limited by aggregation as fluids cool during transport.

At the ionic strengths of these experimental solutions in contact with Grimsel granodiorite and Wyoming bentonite, where in situ ionic strength increases from an initial I ~0.005 M to I > 0.01 M (Supplementary Data, Tables S1–S7), bentonite colloid stability is expected to decrease even at elevated temperatures [76]. Strong decreases in the energy maxima, and generally decreases to colloidal stability, have been noted in experimental studies as evidence that montmorillonite colloids are increasingly more stable in solution with decreasing ionic strengths, especially where I < 0.004 M [76,77]. DLVO theory (Derjaguin-Landau-Verwey-Overbeek, following [80,81]; see also [82,83]) in addition to the experimental literature specific to montmorillonite aggregation shows that at all temperatures and pH, increases in ionic strength decrease the maxima of total interaction energy and promote colloid aggregation rather than suspension in solution. The high ionic strengths of endemic groundwaters in crystalline rock combined with increases to ionic strength during local equilibration with bentonite barriers as observed in our experiments may then inhibit widespread colloidal transport by promoting colloid aggregation.

The sensitivity of the colloids formed in our experiments to ionic strength was additionally observed when the reactant gel from IEBS-5 was dispersed in a more dilute solution. When dispersed in DI water for Zetasizer analysis, the dispersed particles that previously composed the bentonite colloid gel were relatively stable and did not re-aggregate to a gel phase. Changing environmental conditions in the subsurface over time may then impact the stability of colloids formed within the bentonite buffer. Long-term projections of conditions in potential repository sites note that the event of future glaciations may introduce dilute meltwater to the naturally occurring groundwaters in granitic bedrock [84,85], inducing locally dilute solutions and promoting colloid dispersion and potentially transport.

## 5. Conclusions

We conducted a series of hydrothermal experiments reacting Wyoming bentonite and Grimsel granodiorite in a simulated groundwater at elevated temperature and pressure conditions to simulate in situ water-rock interactions during a heating event. The results were interpreted considering the mineralogical evolution that may occur during a heating event in a geological nuclear waste repository sited in a crystalline host rock. These results raise several points that are of interest to how a heating event may affect the mineralogical stability of the EBS in a crystalline repository:

- No montmorillonite-to-illite transition or reduction in the relative abundance of swelling clay was identified by any mineralogical or chemical analyses performed on the reacted bentonite buffer material. Dissolution of potassium-bearing mineral species in the host rock (K-feldspar and micas, Figure 3 and Table 2) was not observed to increase the potassium concentration of the solution, suggesting potential sequestration of $K^+$ through cation exchange in clays. Illitization of montmorillonite in Wyoming bentonite in a Grimsel granodiorite wall rock environment may then be kinetically limited at the experimental conditions.

- CSH minerals were identified locally by SEM and EMP investigations but were not quantitatively identified by QXRD as a constituent of the solid reactants post-mortem. Due to the rapid decrease in calcium early in the experiments and insolubility of calcite in the carbonate-bearing GW at high temperatures, CSH formation is interpreted to have occurred early in the experiments before being retarded by low Ca concentrations.

- Zeolite formation (analcime) was observed in all experiments and was most abundant in the long term (24-week) experiment. Because zeolites were additionally identified in the experiment reacting only bentonite their formation is interpreted to result from reactions involving the alteration of montmorillonite.

- Characteristic secondary mineral assemblages formed at the steel surfaces of all steel types included in the hydrothermal experiments, dominated by an Fe-smectite with minor chlorite and CSH phases. Fe,Ni,Cr sulfides were identified at the surfaces of the reacted stainless steel coupons (316 SS in IEBS-2 and -5; 304 SS in IEBS-3); no such sulfides were identified at the surface of reacted low-carbon steel (IEBS-4).

Additionally, Fe-smectite (as saponite) was not largely identified in bulk mineralogical analyses of the clay matrix, suggesting that Fe-uptake is limited to mineral precipitates at the steel surface.

- Colloid formation observed in our experiments largely formed an aggregated bentonite colloid gel. It is notable that the ionic strength at experimental conditions was relatively high (I ~0.005 to 0.04); when collected and dispersed in DI water the colloids remained dispersed and analyses showed them to be at a relatively stable size. Combined with an increase in bentonite colloid stability at higher temperatures, decreases in the ionic strength of water in contact with a bentonite buffer may increase the potential for colloid-mediated transport over time.

This work reflects the interactions between select EBS materials in a crystalline host rock environment with a dominantly $NaHCO_3$ groundwater chemistry. The results from this study may still highlight reactions that may take place in a more complex system. For example, cementation effects of CSH minerals may then be more extensive in systems with reactive calcium-bearing phases, specifically at cement-bentonite interfaces. Future work may address the extent the observed mineral precipitants influence the engineered barrier performance for arresting the transport of radionuclides through the EBS system or the repository system as a whole.

**Supplementary Materials:** The following supporting information can be downloaded at: https://www.mdpi.com/article/10.3390/min12121556/s1: Tables S1–S7: Aqueous data.

**Author Contributions:** Conceptualization, F.A.C. and E.N.M.; methodology, F.A.C.; investigation, M.R., K.B.S., K.T. and F.A.C.; formal analysis, K.B.S., M.R. and A.Z.; visualization, K.B.S., K.T. and A.Z.; writing—original draft, A.Z. and K.B.S.; writing—review and editing, A.Z.; project administration, F.A.C. and E.N.M.; funding acquisition, E.N.M. All authors have read and agreed to the published version of the manuscript.

**Funding:** This research was funded through the Department of Energy's Spent Fuel and Waste Disposition campaign.

**Data Availability Statement:** The data presented in this study are available in this article and in Supplementary Tables S1–S7 (aqueous data). Additional data will be made available upon request.

**Acknowledgments:** Scanning electron microscopy facilities were provided by the Materials Science and Technology group at Los Alamos National Laboratory. Lindsay Hunt at the University of Oklahoma assisted with the electron microprobe analyses. Rose Harris and Oana Marina at Los Alamos National Laboratory performed the aqueous geochemical analyses. Bentonite Performance Minerals LLC provided the bentonite used in this study. Los Alamos National Laboratory has assigned the free release number LA-UR-22-23154 to this manuscript.

This paper describes objective technical results and analysis. Any subjective views or opinions that might be expressed in the paper do not necessarily represent the views of the U.S. Department of Energy or the United States Government.

This is a technical paper that does not take into account the contractual limitations under the Standard Contract for Disposal of Spent Nuclear Fuel and/or High-Level Radioactive Waste (Standard Contract) (10 CFR Part 961). For example, under the provisions of the Standard Contract, DOE does not consider spent nuclear fuel in multi-assembly canisters to be an acceptable waste form, absent a mutually agreed to contract amendment. To the extent discussions or recommendations in this paper conflict with the provisions of the Standard Contract, the Standard Contract governs the obligations of the parties, and this paper in no manner supersedes, over-rides, or amends the Standard Contract. This paper reflects technical work which could support future decision making by DOE. No inferences should be drawn from this paper regarding future actions by DOE, which are limited both by the terms of the Standard Contract and a lack of Congressional appropriations for the Department to fulfill its obligations under the Nuclear Waste Policy Act including licensing and construction of a spent nuclear fuel repository.

This article has been authored by an employee of National Technology & Engineering Solutions of Sandia, LLC under Contract No. DE-NA0003525 with the U.S. Department of Energy (DOE). The employee owns all right, title and interest in and to the article and is solely responsible for its contents. The United States Government retains and the publisher, by accepting the article for

publication, acknowledges that the United States Government retains a non-exclusive, paid-up, irrevocable, world-wide license to publish or reproduce the published form of this article or allow others to do so, for United States Government purposes. The DOE will provide public access to these results of federally sponsored research in accordance with the DOE Public Access Plan https://www.energy.gov/downloads/doe-public-access-plan (accessed on 20 November 2022).

Sandia National Laboratories is a multimission laboratory managed and operated by National Techology & Engineering Solutions of Sandia, LLC, a wholly owned subsidiary of Honeywell International Inc., for the U.S. Department of Energy's National Nuclear Security Administration under contract DE-NA0003525. Sandia National Laboratories has assigned the number SAND2022-16582 J to the text.

**Conflicts of Interest:** The authors declare no conflict of interest.

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
