# Peer review of "Impacts of Crystalline Host Rock on Repository Barrier Materials at 250 °C: Hydrothermal Co-Alteration of Wyoming Bentonite and Steel in the Presence of Grimsel Granodiorite"

_minerals, doi:10.3390/min12121556_

Round 1

Reviewer 1 Report

The work is devoted to studying the interaction of bentonite safety buffer with steel container, crystalline host rock and groundwater at elevated temperatures. The article presents new, relevant data, and the study itself was carried out using all the necessary analytical methods and is of scientific interest. However, the authors need to finalize the article and answer some questions.

1. Correction of the references.

Line 35-37 - links should be formatted not in the form [1]–[4], [5], [6], but [1-4] [5,6].

Line 181, 511, 515, 540-541, 588, 616, 657-660 - correct in accordance with the rules of the journal.

2. There is no reference in the text to Fig.1.
3. The change in the concentration of potassium in the solution is an important part of the study, as it is discussed throughout the work. The authors draw conclusions based on it. The graph must be added to Fig. 1.
4. Line 316 - quartz is a fairly high-temperature mineral. What is the reason for its increase? With the dissolution of the rest of the mass of minerals? An explanation is required.
5. Clay mineralogy. The diagnosis of sapponite in such small amounts (0.6%) is not a trivial task. To confirm its appearance, additional studies are needed, such as infrared spectroscopy. Or fitting XRD curves with higher resolution.
6. Clay swelling properties. Speaking about the properties of clay minerals, there are free swelling and swelling pressure. These properties can be studied on macrosamples. The study of the swelling properties of smectites based on xrd results is not correct. In this case, we are talking about a change in the structure of smectite and a change in its reflections according to the results of xrd in natural, oriented, and ethylene glycol-saturated samples. An explanation is needed in the text.

Author Response

The authors appreciate the constructive comments and have carefully considered the raised concerns. In effort to address those concerns the following changes were made:

1. Correction of the references.

The suggested changes have been made.

2. There is no reference in the text to Fig.1.

Figure 1 and subplots are referenced now referenced at relevant places throughout the text, especially in the results section.

3. The change in the concentration of potassium in the solution is an important part of the study, as it is discussed throughout the work. The authors draw conclusions based on it. The graph must be added to Fig. 1

The authors appreciate the suggestion – Fig. 1 has been adapted to include K (and Na) concentrations.

4. Line 316 - quartz is a fairly high-temperature mineral. What is the reason for its increase? With the dissolution of the rest of the mass of minerals? An explanation is required.

The relative increase in quartz as a fraction of the analyzed sample was interpreted to be due to the dissolution (and relative decrease as a fraction of the whole) of more reactive mineral phases in the sample. The reporting approach is made explicit in the Methods section 2.2 (at line 228) and more explicitly addressed in the Results section 3.2 (starting at line 308 of the revised manuscript).

5. Clay mineralogy. The diagnosis of sapponite in such small amounts (0.6%) is not a trivial task. To confirm its appearance, additional studies are needed, such as infrared spectroscopy. Or fitting XRD curves with higher resolution.

On recalculating the full pattern fitting for the samples, no unique peaks that could be attributed to saponite were identified. As the estimated % of the sample (0.6) was well under the estimated error for this method (+- 5 wt % for clay minerals), and in the absence of corroborating identification methods, saponite is now excluded from the reported calculation.

6. Clay swelling properties. Speaking about the properties of clay minerals, there are free swelling and swelling pressure. These properties can be studied on macrosamples. The study of the swelling properties of smectites based on xrd results is not correct. In this case, we are talking about a change in the structure of smectite and a change in its reflections according to the results of xrd in natural, oriented, and ethylene glycol-saturated samples. An explanation is needed in the text.

The authors appreciate the clarity on this point – the vocabulary has been changed to be more precise throughout the text, and especially in Methods section 2.2 (lines 235-238).

Reviewer 2 Report

For clarity, it would be useful to supplement the output from PHREEQC with a graph showing the weight or percentage of the attached phase in solution as a function of the pH of the solution.

What led you to use the HCO3-concentration of 0.02 mol L–1 in your calculations. Is this a calculation or has the concentration been determined?

Author Response

The authors appreciate the thoughtful review and have made the following changes responsive to the raised concerns:

  1. For clarity, it would be useful to supplement the output from PHREEQC with a graph showing the weight or percentage of the attached phase in solution as a function of the pH of the solution.

The authors appreciate the insight and suggestion. Unfortunately, thermodynamic data for the phases of interest to these experiments as presented in Fig. 6 (specifically CSH phases and zeolite minerals) are limited at the experimental conditions. In addition, our experimental results yielded no quantitative data on amounts of precipitated phases of interest in our experiments that could be compared to such a graph, especially since the precipitation of apparently undersaturated minerals (see for example analcime) suggests that the system has not reached a thermodynamic equilibrium within the experimental time. For these reasons, we elected not to include a figure reflecting the suggestion and maintained the discussion to the compartive saturation indices (Q/K) presented in Fig. 6; the effect of pH and limitations of the thermodynamic data are noted in the Discussion section 4.3 (see especially lines 559-567).

  1. What led you to use the HCO3-concentration of 0.02 mol L–1 in your calculations. Is this a calculation or has the concentration been determined?

The concentration of HCO3- used in the speciation calculations was based on the initial NaHCO3 content added to the initial solution – no carbonate measurements of the solution were performed. A comment to this effect is now included in the Methods section 2.2 (lines 217-218).